# Epitranscriptomic $N^6$-Methyladenosine Profile of SARS-CoV-2-Infected Human Lung Epithelial Cells

Stacia Phillips,[a] Tarun Mishra,[a] Shaubhagya Khadka,[a] Dana Bohan,[a*] Constanza E. Espada,[a] Wendy Maury,[a] Li Wu[a]

[a]Department of Microbiology and Immunology, Carver College of Medicine, University of Iowa, Iowa City, Iowa, USA

**ABSTRACT** $N^6$-methyladenosine (m$^6$A) is a dynamic posttranscriptional RNA modification that plays an important role in determining transcript fate. The functional consequence of m$^6$A deposition is dictated by a group of host proteins that specifically recognize and bind the m$^6$A modification, leading to changes in RNA stability, transport, splicing, or translation. The cellular m$^6$A methylome undergoes changes during certain pathogenic conditions such as viral infections. However, how m$^6$A modification of host cell transcripts and noncoding RNAs change during severe acute respiratory syndrome coronavirus (SARS-CoV-2) infection has not been reported. Here, we define the epitranscriptomic m$^6$A profile of SARS-CoV-2-infected human lung epithelial cells compared to uninfected controls. We identified mRNA and long and small noncoding RNA species that are differentially m$^6$A modified in response to SARS-CoV-2 infection. The most significantly differentially methylated transcript was the precursor of microRNA-4486 (miRNA-4486), which showed significant increases in abundance and percentage of methylated transcripts in infected cells. Pathway analyses revealed that differentially methylated transcripts were significantly associated with several cancer-related pathways, protein processing in the endoplasmic reticulum, cell death, and proliferation. Upstream regulators predicted to be associated with the proteins encoded by differentially methylated mRNAs include several proteins involved in the type-I interferon response, inflammation, and cytokine signaling.

**IMPORTANCE** Posttranscriptional modification of viral and cellular RNA by $N^6$-methyladenosine (m$^6$A) plays an important role in regulating the replication of many viruses and the cellular immune response to infection. We therefore sought to define the epitranscriptomic m$^6$A profile of human lung epithelial cells infected with SARS-CoV-2. Our analyses demonstrate the differential methylation of both coding and noncoding cellular RNAs in SARS-CoV-2-infected cells compared to uninfected controls. Pathway analyses revealed that several of these RNAs may be involved in the cellular response to infection, such as type-I interferon. Our study implicates m$^6$A modification of infected-cell RNA as a mechanism of posttranscriptional gene regulation during SARS-CoV-2 infection.

**KEYWORDS** $N^6$-methyladenosine, SARS-CoV-2, epitranscriptomics, infection, lung epithelial cells, microarray

Address correspondence to Li Wu, li-wu@uiowa.edu.

*Present address: Dana Bohan, Vir Biotechnology, Inc., San Francisco, California, USA.

The authors declare no conflict of interest.

$N^6$-methyladenosine (m$^6$A) is the most prevalent posttranscriptional modification of eukaryotic mRNA and plays an important role in the fate of the modified mRNA molecule. The m$^6$A is deposited on adenosine by a methyltransferase, or "writer," complex consisting of the catalytic heterodimer methyltransferase-like 3 and methyltransferase-like 14 (METTL3/METTL14) in complex with the adapter protein Wilms' tumor 1-associated protein (WTAP) (1). m$^6$A is also prevalent on small noncoding RNA (sncRNA) and long noncoding RNA (lncRNA) and this modification is catalyzed by the writer METTL16 (2). The two demethylases, or "erasers," fat mass and obesity-associated protein (FTO) and $\alpha$-ketoglutarate-dependent dioxygenase AlkB homolog 5 (ALKBH5) can remove the

m⁶A modification, suggesting that m⁶A modification is not only dynamic but reversible (3, 4). The outcome of m⁶A modification is dictated by m⁶A-specific RNA binding proteins or "readers," the most well characterized of which are members of the YT521-B homology (YTH) family (5, 6). The binding of readers to the modified mRNA can lead to changes in stability, translation, localization, and splicing (7–12). Therefore, m⁶A modification acts as an important mechanism of posttranscriptional regulation of gene expression.

Many virus genomes and viral RNAs are m⁶A modified, and these modifications play important functional roles in various stages of virus replication and evasion of innate immune sensing (13). Severe acute respiratory syndrome coronavirus (SARS-CoV-2) RNAs are m⁶A modified, and while some cell type-dependent discrepancies exist, most studies have reported that m⁶A is required for efficient virus replication (14–18). In addition to functional m⁶A modification of viral RNAs, changes in the cellular m⁶A methylome have also been shown to occur in association with viral infections (19–21). Of particular interest, cellular transcripts involved in establishing an antiviral immune response are posttranscriptionally regulated by m⁶A modification (22–24). It is likely that SARS-CoV-2 infection leads to changes in the m⁶A modification state of host cell transcripts, either induced directly by the virus or through the cellular response to infection. Indeed, m⁶A sequencing (m⁶A-seq or meRIP-seq) has revealed the loss or gain of m⁶A modifications in host cell RNA from infected cells (14). However, how the host m⁶A methylome changes in human lung cells in response to SARS-CoV-2 infection remains unknown.

Here, we report the results of epitranscriptomic m⁶A microarray analysis of human lung cells infected with SARS-CoV-2 compared to uninfected control cells. We identified changes in the abundance of methylated cellular RNAs for both protein-coding and noncoding transcripts. One micro-RNA (miR) precursor, miR-4486, was found to be 175 times more abundant in the methylated fraction of infected-cell RNA compared to uninfected controls. Interestingly, biological pathway analysis revealed that many differentially methylated mRNA transcripts code for proteins that are regulated upstream by proteins involved in inflammation, cytokine signaling, and innate immunity. These findings will serve as the basis for future functional validation studies to determine how changes in the methylation status of host cell transcripts may affect SARS-CoV-2 replication and viral pathogenesis.

## RESULTS

**A549-hACE2 cells support productive SARS-CoV-2 infection.** We sought to determine the epitranscriptomic m⁶A profile of SARS-CoV-2-infected cell RNA using a human lung epithelial cell line, as lung epithelial cells represent a biologically relevant target of SARS-CoV-2. Robust and reliable identification of changes to the methylation level of individual host cell transcripts during infection is best achieved using conditions under which most of the cells have become infected, reducing the background signal contributed by uninfected cells. Therefore, we first directly compared three different lung cell lines (A549-hACE2 cells expressing human angiotensin-converting enzyme 2 [hACE2], Calu-3, and H1650) for their ability to support SARS-CoV-2 replication under identical conditions. In our infection assays, we chose to infect cells with SARS-CoV-2 (strain USA-WA-1/2020) at a multiplicity of infection (MOI) of 1 for 24 h to allow for a full viral life cycle and the spreading of infection to occur (25). After 24 h, A549-hACE2 cells were fixed, and infected cells were visualized by immunofluorescent staining using a SARS-CoV-2 nucleocapsid-specific antibody (Fig. 1A). Infection of A549-hACE2 cells resulted in a greater proportion of N-positive cells (~70%) compared to Calu-3 and H1650 cell lines (data not shown). Real-time quantitative PCR (RT-qPCR) analysis using *spike* gene-specific qPCR primers demonstrated robust viral RNA replication in A549-hACE2 cells with ~5 × 10⁴ copies of spike RNA present per infected cell (Fig. 1B). These RNA molecules represent both full-length positive sense RNA and subgenomic RNA used for translation to viral protein. Based on these results, we chose to use A549-hACE2 cells to determine the epitranscriptomic m⁶A profile of SARS-CoV-2-infected cells.

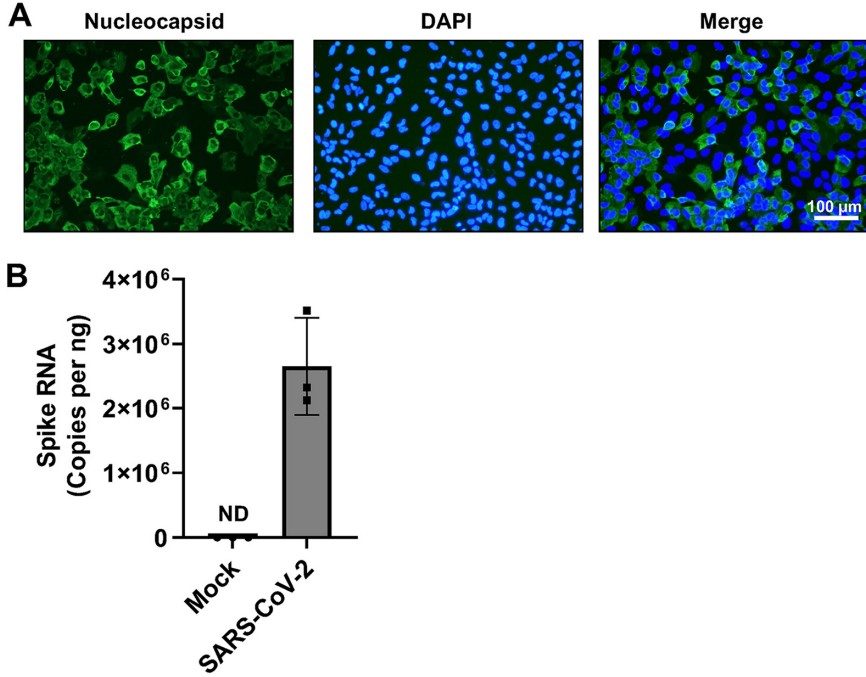

**FIG 1** SARS-CoV-2 infection of A549-hACE2 cells. A549-hACE2 cells were infected with SARS-CoV-2 (strain USA-WA1/2020) at an MOI of 1 for 24 h. (A) Immunofluorescent staining was performed to visualize infected cells by the presence of SARS-CoV-2 nucleocapsid (green). Nuclei of cells are stained with DAPI (blue). (B) SARS-CoV-2 spike RNA in infected cells ($n = 3$, biological triplicate) was quantified by RT-qPCR. ND, not detected.

**SARS-CoV-2 infection of A549-hACE2 cells leads to differential m⁶A modification of cellular RNA.** To analyze m⁶A modifications of host cell transcripts, A549-hACE2 cells were infected with SARS-CoV-2 at an MOI of 1 for 24 h in biological triplicate. Total RNA from SARS-CoV-2-infected cells and mock-infected negative-control cells was used for m⁶A immunoprecipitation (IP) followed by microarray analysis using the Arraystar Epitranscriptomic m⁶A Array (see schematic method summary in Fig. 2A). The RNA present in the IP fraction represents m⁶A-modified RNA, whereas the remaining unbound fraction is assumed to be unmethylated. Each fraction was fluorescently labeled and mixed prior to array hybridization. The microarray contains over 60,000 unique probes (60 nt each) that represent 44,122 mRNA, 12,496 lncRNA, 1,366 pre-miRNA, 1,642 pri-miRNA, 19 small nuclear RNA (snRNA), and 786 small nucleolar RNA (snoRNA) transcripts. Unique mRNA splice isoforms are distinguished by probes that are exon specific or span a splice junction. The signal for each transcript in the IP and unbound fractions was normalized to the intensity of non-human spike-in RNA.

Transcript types that were found to be significantly differentially methylated ≥1.5-fold with $P \leq 0.05$ in SARS-CoV-2-infected cells compared to mock-infected controls are summarized in Table 1. A total of 186 unique transcripts were hypomethylated and 119 transcripts were hypermethylated in response to infection. A volcano plot shows the statistical significance ($-\log_{10} P$ value) versus the fold change ($\log_2$ FC) in the abundance of methylated transcript in SARS-CoV-2-infected cells relative to mock (Fig. 2B). A selection of transcripts with the most significant changes in m⁶A abundance ($P \leq 0.005$) is shown in Table 2. The full list of transcripts with significantly different m⁶A abundance in infected cells can be found in Table S1 in the supplemental material ($P \leq 0.05$; fold change ≥1.5).

The plot of $\log_2$ FC shows that changes in hypomethylated transcripts ranged from a $\log_2$ FC of −0.59 to −1.52 (1.5-fold to 2.8-fold change). Similarly, the $\log_2$ FC for most transcripts found to be hypermethylated in infected cells ranged from 0.59 to 1.47 (1.5-fold to 2.8-fold change). Eight of the hypermethylated transcripts showed a greater than 3-fold

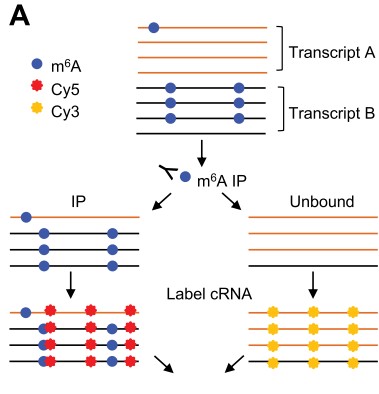

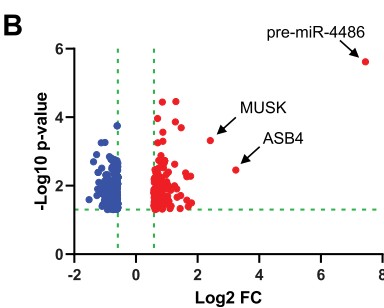

**FIG 2** Epitranscriptomic m⁶A microarray of SARS-CoV-2-infected A549-hACE2 cells. (A) Schematic overview of the method. Total cellular RNA from each sample (SARS-CoV-2-infected and mock-infected controls, biological triplicate, $n = 3$ each group) was used for immunoprecipitation using an m⁶A-specific antibody. Methylated and unmethylated RNA fractions were fluorescently labeled (Cy3 or Cy5) prior to array hybridization (refer to Materials and Methods for details). (B) Volcano plot of transcripts containing higher (red) and lower (blue) levels of m⁶A modification in infected cells compared to mock-infected control cells. The miRNA precursor (pre-mir-4486) with the most significant m⁶A change is labeled.

change. Remarkably, the primary miR transcript for miR-4486 was found in the methylated RNA fraction from infected cells at a 175-fold higher level than in uninfected controls. The associated $P$ value of this change was $2.41 \times 10^{-6}$, demonstrating high reproducibility among three independent infections and uninfected controls (Fig. 2B and Table S1).

A unique feature of the epitranscriptomic microarray is the ability to determine the percentage of transcript molecules that are m⁶A modified, based on the relative intensity of signals in the IP and unbound fractions. This stoichiometric information is not provided by other m⁶A detection techniques such as m⁶A-seq (26, 27). Selected transcripts with a significantly different percentage of m⁶A-modified RNA in SARS-CoV-2-infected samples relative to mock controls are shown in Table 3 ($P$ value $\leq 0.005$). The full list of transcripts with a significantly different percentage of m⁶A-modified RNA in infected cells can be found in Table S1 ($P \leq 0.05$). These results show that many cellular transcripts undergo changes in m⁶A abundance and percentage of transcript modified in response to SARS-CoV-2 infection of A549-hACE2 cells.

The methylation status of selected transcripts was validated using m⁶A-RNA IP (RIP) followed by qPCR (Fig. 3A). A549-hACE2 cells were infected with a VSV-G-pseudotyped

**TABLE 1** Summary of differentially modified transcripts by type in SARS-CoV-2-infected A549-hACE2 cells versus mock-infected control cells[a]

| Transcript type | Hypo m⁶A | Hyper m⁶A | Total | No. of probes |
|---|---|---|---|---|
| mRNA | 160 | 89 | 249 | 44,122 |
| lncRNA | 24 | 22 | 46 | 12,496 |
| sncRNA | 2 | 8 | 10 | 3,813 |

[a]Fold change $\geq 1.5$ and $P \leq 0.05$.

**TABLE 2** Selected differentially m⁶A-modified transcripts in SARS-CoV-2-infected A549-hACE2 cells versus mock-infected control cells by m⁶A quantity[a]

| Transcript type | Transcript ID | Gene symbol | Log2-fold change | P value |
|---|---|---|---|---|
| mRNA | ENST00000250457 | EGLN3 | 0.8595436 | 3.609E-05 |
| mRNA | HBMT00001402195 | CATG00000101027.1 | 1.4687341 | 0.0002032 |
| mRNA | ENST00000540159 | BNIP3 | 0.8732292 | 0.0002728 |
| mRNA | ENST00000325885 | ASB4 | 2.4125726 | 0.0004796 |
| mRNA | ENST00000618819 | PTPN9 | 0.8113196 | 0.0013265 |
| mRNA | NM_001330464 | ATL2 | 0.9303607 | 0.0018649 |
| mRNA | ENST00000357077 | ANK2 | 0.8673231 | 0.0021318 |
| mRNA | ENST00000422053 | TRIB3 | 0.9064073 | 0.0021784 |
| mRNA | ENST00000528331 | SYBU | 1.255638 | 0.0023711 |
| mRNA | ENST00000311450 | PLAC8L1 | 0.8936838 | 0.0029927 |
| mRNA | ENST00000382181 | RBCK1 | 0.7350803 | 0.0030862 |
| mRNA | ENST00000248071 | KLF2 | 0.6192452 | 0.0034099 |
| mRNA | ENST00000374448 | MUSK | 3.2415767 | 0.0034892 |
| mRNA | ENST00000376468 | NPPB | 1.0536428 | 0.0040088 |
| mRNA | ENST00000371445 | DMRTB1 | 0.6458079 | 0.0041402 |
| mRNA | ENST00000644112 | TIMM8A | −0.6091437 | 0.0001784 |
| mRNA | ENST00000360948 | NTRK3 | −1.0030983 | 0.0005479 |
| mRNA | ENST00000629765 | NTRK3 | −1.1272045 | 0.0005579 |
| mRNA | ENST00000394480 | NTRK3 | −1.2754039 | 0.0012382 |
| mRNA | ENST00000417456 | RP11-244H3.5 | −0.8596983 | 0.0014264 |
| mRNA | ENST00000619416 | KIAA0586 | −0.7813802 | 0.0015997 |
| mRNA | ENST00000539097 | HIF1A | −0.8053586 | 0.0017167 |
| mRNA | ENST00000451528 | ST8SIA4 | −0.6902188 | 0.0017212 |
| mRNA | NM_001320134 | NTRK3 | −0.61504 | 0.0018528 |
| mRNA | ENST00000378133 | PCDHA1 | −0.5856729 | 0.0022007 |
| mRNA | ENST00000333896 | SPTBN1 | −0.7070035 | 0.002238 |
| mRNA | ENST00000368581 | RSPH4A | −0.6288855 | 0.0026367 |
| mRNA | ENST00000394332 | RPL23 | −0.6408456 | 0.0026379 |
| mRNA | ENST00000357496 | QRICH1 | −0.593256 | 0.0027733 |
| mRNA | ENST00000537259 | SLC24A1 | −0.7748985 | 0.0028879 |
| mRNA | MICT00000118072 | CATG00000023300.1 | −0.6383486 | 0.0029291 |
| mRNA | ENST00000644112 | TIMM8A | −0.6091437 | 0.0001784 |
| mRNA | ENST00000360948 | NTRK3 | −1.0030983 | 0.0005479 |
| lncRNA | NR_148507 | AC010982.1 | 1.2975579 | 3.476E-05 |
| lncRNA | ENST00000477817 | PKP1 | 0.7016014 | 0.0001088 |
| lncRNA | NR_152836 | PSORS1C3 | 1.2784829 | 0.0001375 |
| lncRNA | ENST00000579368 | RP11-674N23.1 | 0.8775907 | 0.0005063 |
| lncRNA | ENST00000592680 | AC007773.2 | 0.7066193 | 0.0005585 |
| lncRNA | ENST00000436551 | AC104654.2 | −1.3765495 | 0.0020037 |
| lncRNA | NR_136184 | CENPO | −0.7059265 | 0.0026148 |
| lncRNA | ENST00000503936 | TTC33 | −0.6501871 | 0.0027399 |
| lncRNA | ENST00000422847 | RP11-40F6.2 | −0.6267391 | 0.0050007 |
| pre-miRNA | MI0001446 | hsa-mir-4486 | 7.4490146 | 2.413E-06 |

[a]$P \leq 0.005$.

single-cycle SARS-CoV-2 replicon [ΔS-VRP(G)], in which the viral spike ORF is replaced with a dual reporter expressing luciferase and mNeonGreen (28). Total RNA was harvested from ΔS-VRP(G)-infected cells and mock-infected negative-control cells at 24 h postransduction and subjected to m⁶A-RIP. RNA in the IP fraction was used for cDNA synthesis, and qPCR was performed for individual transcripts (Fig. 3B). The same approach was used to validate increased methylation of miR-4486 in the presence of ΔS-VRP(G). While total levels of miR-4486 remained unchanged, there was a significant enrichment of miR-4486 in the m⁶A-RIP (Fig. 3C and D). In addition to validating the results of our epitranscriptomic array, these data suggest that the differential methylation of these particular transcripts is independent of SARS-CoV-2 spike.

**Biological pathway analysis of differentially methylated protein-coding transcripts.** Protein coding transcripts were analyzed by iPathwayGuide (Advaita) to identify biological pathways that are significantly associated with cellular mRNAs that are differentially methylated in response to SARS-CoV-2 infection of A549-hACE2 cells. The P values for significance of the association are derived from a combination of two independent

**TABLE 3** Selected differentially m⁶A-modified transcripts in SARS-CoV-2-infected A549-hACE2 cells versus mock-infected control cells by percent modified[a]

| Transcript type | Transcript ID | Gene symbol | Percent mock | Percent SARS-CoV-2 |
|---|---|---|---|---|
| mRNA | ENST00000374448 | MUSK | 30% | 69% |
| mRNA | ENST00000371968 | LHFPL1 | 33% | 43% |
| mRNA | ENST00000264657 | STAT3 | 27% | 34% |
| mRNA | ENST00000528331 | SYBU | 40% | 53% |
| mRNA | ENST00000360375 | LRRCC1 | 40% | 23% |
| mRNA | ENST00000361987 | CNTF | 91% | 81% |
| mRNA | ENST00000216513 | SIX4 | 93% | 82% |
| mRNA | ENST00000367512 | EDEM3 | 83% | 69% |
| mRNA | ENST00000522232 | CTB-83D3.1 | 72% | 34% |
| mRNA | ENST00000361028 | ZSCAN12 | 80% | 54% |
| lncRNA | NR_145484 | ARHGAP32 | 91% | 74% |
| lncRNA | NR_015352 | CECR7 | 65% | 32% |
| pre-miRNA | MI0001446 | hsa-mir-4486 | 44% | 73% |

[a]$P \leq 0.005$.

analyses, classical overrepresentation analysis (pORA) and a measure of accumulated perturbation of a given pathway (pAcc) (29–32). The full list of pathways associated with at least one differentially methylated mRNA is listed in Table S2, with pAcc, pORA, and the combined $P$ value indicated. Pathways with no pAcc represent metabolic networks, as opposed to signaling pathways that have both pAcc and pORA.

Figure 4 shows the list of pathways with the highest combined significance ($P \leq 0.005$). We found that protein-coding transcripts that are differentially methylated in response to SARS-CoV-2 infection are associated with several cancer-related pathways (microRNAs, pathways, proteoglycans, and programmed death ligand 1 [PD-L1] expression and PD-1 checkpoint pathways in cancer); infectious disease (Legionellosis, Kaposi sarcoma-associated herpesvirus infection, and hepatitis B); cell metabolism, proliferation, and survival/death (protein processing in the endoplasmic reticulum, metabolic pathways, necroptosis, forkhead box O [FoxO] signaling, mitophagy, epidermal growth factor receptor [EGFR] tyrosine kinase inhibitor resistance, signaling pathways regulating pluripotency of stem cells, phosphoinositide-3-kinase-Akt [PI3K-Akt] signaling); and the immune response (JAK-STAT signaling). Our data also indicated the number of differentially methylated transcripts that are associated with each pathway (count) and the $-\log_{10}$ of the associated combined $P$ value. Together, these results suggest complex and dynamic biological pathways are involved in cellular responses to SARS-CoV-2 infection.

**Upstream regulators of differentially methylated mRNAs and predicated networks.** To further analyze the significant regulators of differentially methylated transcripts during SARS-CoV-2 infection, iPathwayGuide was also used to identify putative upstream regulators of proteins encoded by transcripts found to be differentially methylated in SARS-CoV-2-infected cells compared to mock-infected control cells. Interestingly, the top 20 upstream regulators predicted with the highest significance ($P \leq 0.01$) were enriched for proteins involved in inflammation, cytokine signaling, and innate immunity (Fig. 5). The most significantly associated predicted upstream regulator is EGFR, which has been implicated as a potential therapeutic target for COVID-19 treatment (33–36). Other predicted upstream regulators with known function in inflammation and innate immunity include mitogen-activated protein kinase kinase 7 (MAP2K7), tumor necrosis factor receptor superfamily member 1A (TNFRSF1A), sprouty RTK signaling antagonist 4 (SPRY4), Janus kinase 3 (JAK3), Janus kinase 2 (JAK2), interferon alpha 1 (IFNA1), and tumor necrosis factor (TNF).

To better understand the interactions among significant regulators, predicted upstream regulators of differentially methylated transcripts were used to construct networks illustrating known regulatory interactions among individual nodes of a given pathway. Selected upstream regulators and their downstream targets that are differentially methylated in response to SARS-CoV-2 infection are indicated, with

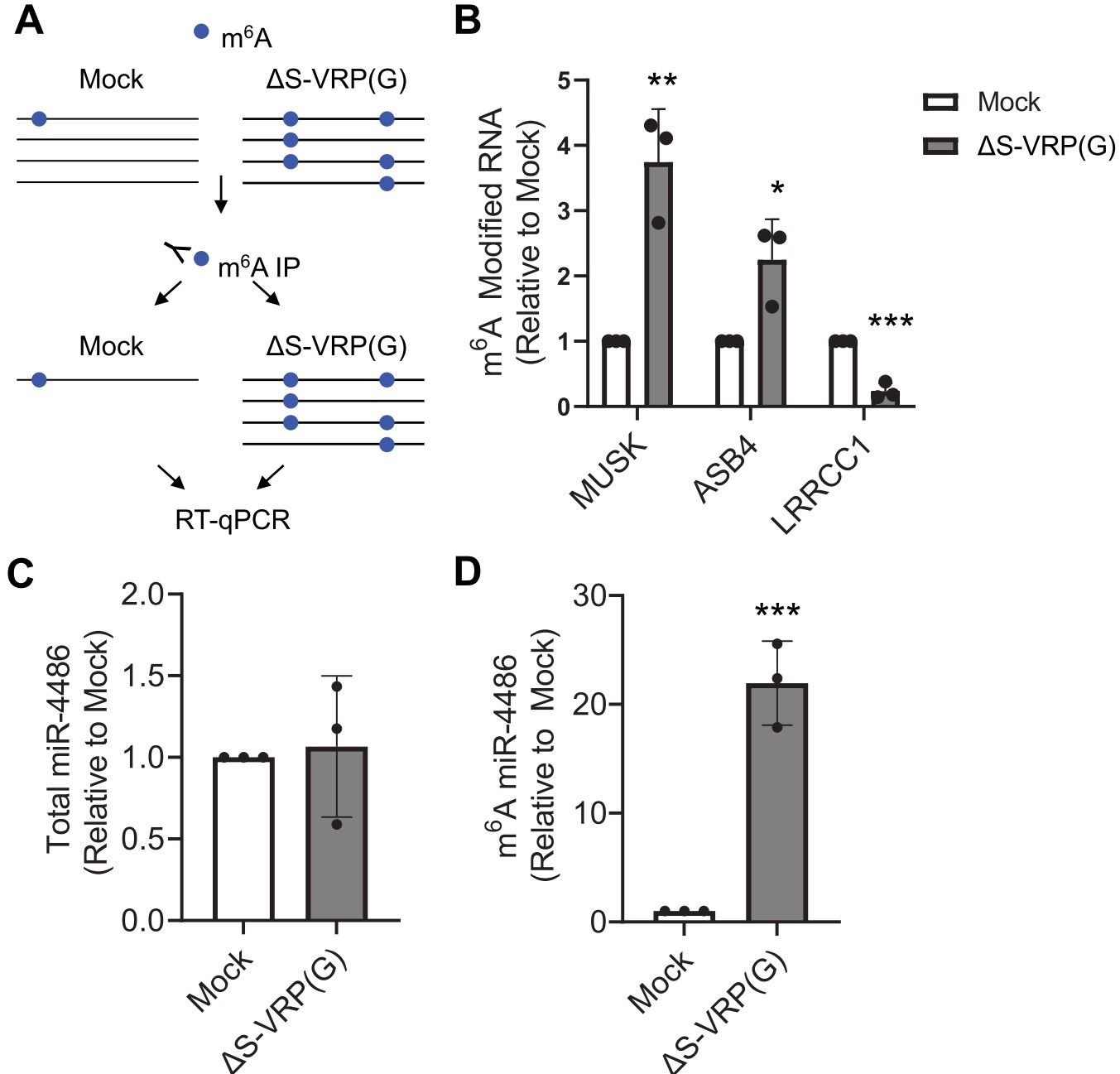

**FIG 3** Validation of differential methylation for selected transcripts in A549-hACE2 cells. (A) Schematic of m⁶A RIP. A549-hACE2 cells were mock infected or infected with ΔS-VRP(G) for 24 h. Total RNA was used for m6A-RIP. (B) Total RNA from cells infected with ΔS-VRP(G) was used for m⁶A-RIP. The fold enrichment of m⁶A-modfied RNA in the IP fraction of infected cell RNA versus mock-infected controls is indicated. (C) Total cellular RNA from mock-infected or ΔS-VRP(G)-infected cells was used for RT-qPCR analysis of total levels of mature miR-4486. (D) Total cellular RNA from mock-infected or ΔS-VRP (G)-infected cells was used for m⁶A RNA-IP to determine the levels of m⁶A-modified mature miR-4486. (C and D) Statistical significance was determined by $t$ test compared to mock control. *, $P < 0.05$; **, $P < 0.005$; ***, $P < 0.0005$.

hypomethylated transcripts shown in blue and hypermethylated transcripts in pink (Fig. 6A to C). Gray nodes represent intermediate genes in the pathway that directly regulate or are regulated by differentially methylated pathway members but do not themselves exhibit any change in methylation status upon SARS-CoV-2 infection. Pink arrows illustrate activation and gray bars represent inhibition for functional interactions that have been experimentally validated (Fig. 6A to C). These analyses allow us to develop novel hypotheses regarding how the host cell responds to SARS-CoV-2 infection. Together, these results suggest that regulation of gene expression at the level of

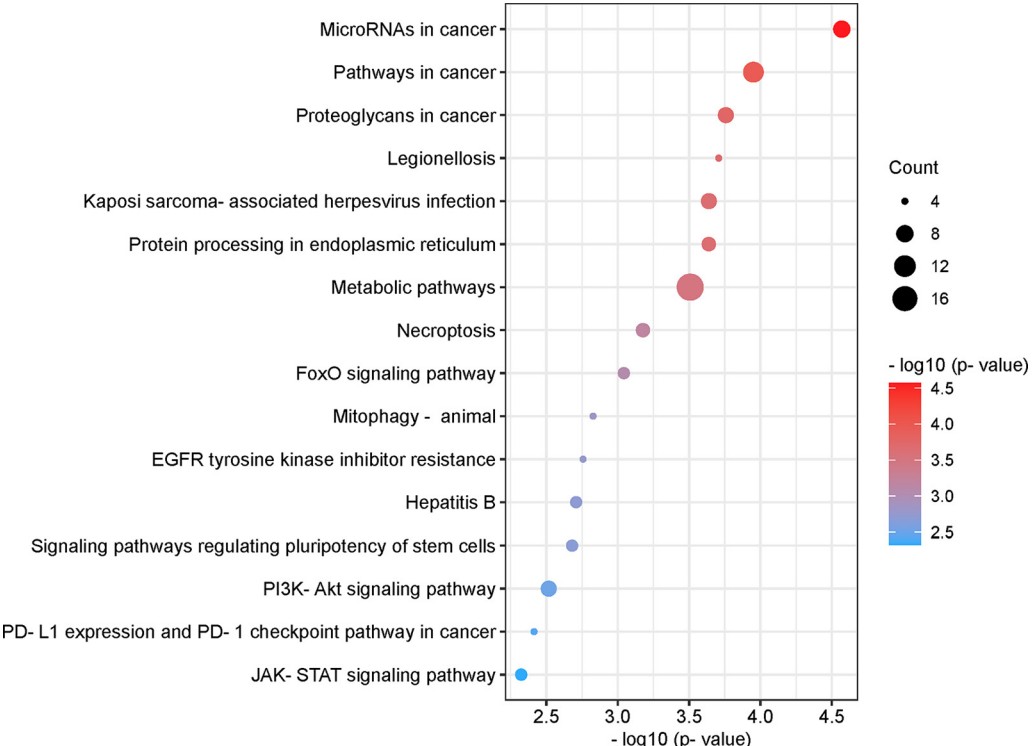

**FIG 4** Pathway analysis of differentially methylated mRNAs. List of pathways associated with differentially methylated mRNAs with the highest combined significance (pORA and pAcc ≤0.005). Significance is indicated on the x axis and by sphere color. The size of the sphere corresponds to the number of differentially methylated mRNAs associated with each pathway (count). Bubble plot was created using https://www.bioinformatics.com.cn/en, a free online platform for data analysis and visualization.

posttranscriptional RNA modification is a mechanism by which the cell responds to SARS-CoV-2 infection and may have effects on viral pathogenesis and the immune response.

## DISCUSSION

Posttranscriptional m⁶A modification of RNA is an important strategy for the regulation of gene expression (37). We sought to identify changes in m⁶A modification of cellular RNA during SARS-CoV-2 infection of human lung epithelial cells using epitranscriptomic m⁶A microarray analysis. These cellular RNAs may be important for virus replication or for establishing an antiviral innate immune response. We identified mRNA and long and small noncoding RNA species that are differentially m⁶A modified in response to SARS-CoV-2 infection. Differentially methylated mRNA transcripts were found to be associated with biological pathways and upstream regulators that are involved in the immune response to viral infection. These data may provide a basis for novel hypotheses regarding mechanisms of SARS-CoV-2 replication or the cellular response to the infection of lung epithelial cells. Future functional studies of the identified cellular RNA are required to test these hypotheses.

Overall, we observed differential methylation of 305 unique transcripts, with 186 hypomethylated and 119 hypermethylated transcripts in infected cells compared to uninfected controls. These differentially methylated transcripts are a relatively small percentage of all transcripts represented on the epitranscriptomic microarray (Table 1). The number of transcripts and magnitude of differential methylation in the current study are likely an underestimation of the actual change in methylation status that occurs in infected cells, due to the contribution of RNA from the ~30% of cells in infected cultures that remained uninfected when the RNA was harvested (Fig. 1A). Some effects may also be missed due to differences in IP efficiency of individual

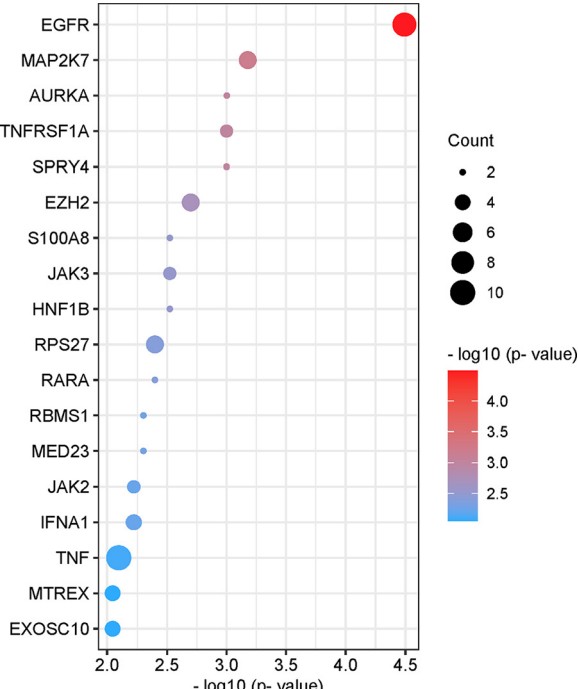

**FIG 5** Upstream regulators of differentially methylated mRNAs. List of upstream regulators associated with differentially methylated mRNAs with the highest significance ($P \leq 0.01$). Significance is indicated on the *x* axis and by sphere color. The size of the sphere corresponds to the number of differentially methylated mRNAs associated with each pathway (count). Bubble plot was created using https://www.bioinformatics.com.cn/en, a free online platform for data analysis and visualization.

transcripts, due either to RNA length or location of the m⁶A modification(s). Although the consensus RRACH motif (R = A or G; H = A, C, or U) for m⁶A modification is relatively common, only ~5% of these motifs are m⁶A modified, resulting in an average of one to two m⁶A modifications per mRNA transcript (26). Therefore, despite the limitations described above, it is possible that modest changes in m⁶A modification can potentiate changes to RNA function during SARS-CoV-2 infection.

One mRNA found to be significantly more m⁶A modified in infected cells compared to mock-infected controls is the muscle-associated receptor tyrosine kinase (MUSK). The epitranscriptomic microarray indicated that this mRNA is present at ~8-fold greater abundance in the m⁶A-RIP fraction in infected cells and that the percentage of methylated transcript increased from 30% in uninfected cells to 69% in SARS-CoV-2-infected cells (Tables 2 and 3). MUSK is a receptor tyrosine kinase that is essential for the formation and maintenance of the neuromuscular junction and is expressed at very low levels in the lung under normal conditions (38). Autoantibodies directed against MUSK inhibit acetylcholine receptor clustering at the neuromuscular junction and are associated with a rare form of myasthenia gravis (MG), a chronic autoimmune disorder in which antibodies destroy the communication between nerves and muscle, resulting in weakness of the skeletal muscles (39). Interestingly, a recent case report identified the development of MUSK-associated MG potentially triggered by SARS-CoV-2 infection (40). It would be important to investigate how SARS-CoV-2 infection could induce the development of autoantibodies to MUSK or how this might involve posttranscriptional m⁶A modifications of the MUSK mRNA.

We also observed a modest but significant increase in the levels of m⁶A modification of signal transducer and activator of transcription-3 (STAT3) transcript (Table 3). This transcription factor plays a pivotal role in intracellular signaling and subsequent activation of gene expression in response to a variety of cytokines and chemokines, including IL-6 and type-I interferons (41). STAT3 may contribute to the pathogenesis of

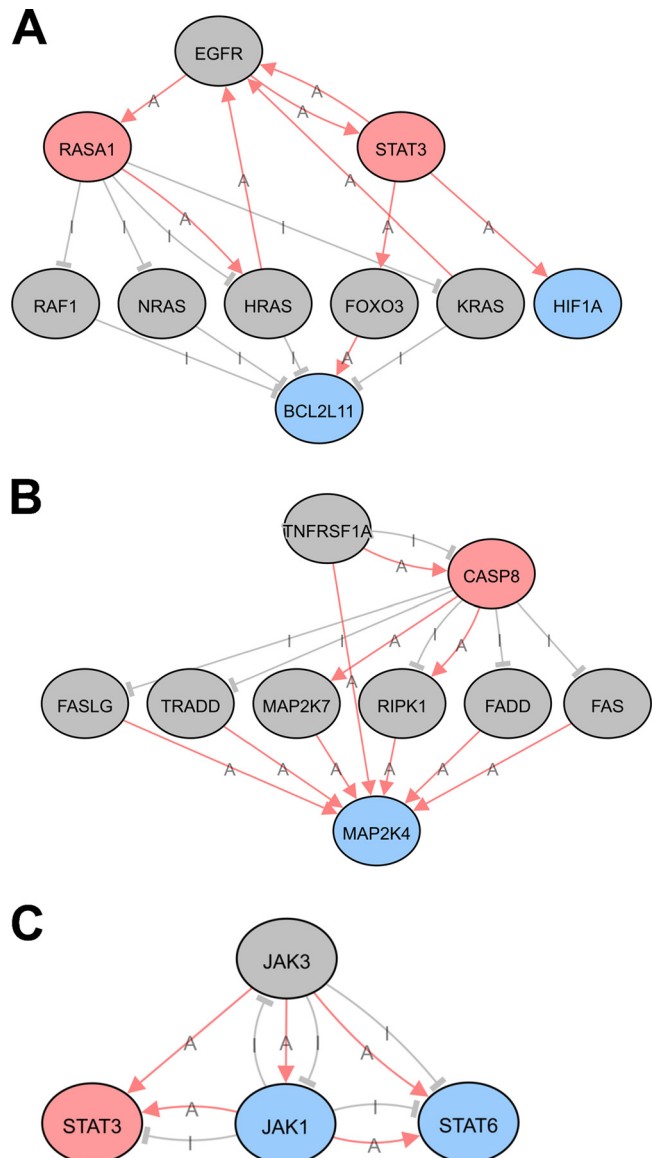

**FIG 6** Predicted upstream regulator networks. Network map of selected upstream regulators of differentially methylated mRNA transcripts. Upstream regulators are EGFR (A), TNFRSF1A (B), and JAK3 (C). Colors represent the change in m⁶A modification of indicated transcripts in response to SARS-CoV-2 infection in human lung epithelial cells. Gray circles: no change; pink circles: hypermethylated; blue circles: hypomethylated. Lines indicate known functional interactions between pathway nodes. Pink arrows, activation (A); gray bars, inhibition (I). Images were obtained using iPathwayGuide from AdvaitaBio.

SARS-CoV-2 infection in a variety of ways considering its pleiotropic effects on inflammation and the immune response (reviewed in reference 42). We also observed changes in the abundance of m⁶A-modified JAK1 and STAT6 transcripts, which are also part of the JAK/STAT signaling pathway and may functionally interact with each other as part of the host cell response to infection (Table S1 and Fig. 6C). Consistent with these results, pathway analysis revealed that the JAK-STAT pathway was significantly associated with differentially methylated mRNAs in infected cells (Fig. 4). Therefore, other members of the JAK-STAT pathway were identified as upstream regulators of the protein products of mRNAs found to be differentially methylated in infected cells (Fig. 5 and 6A and C). This overrepresentation of the JAK-STAT pathway in our analyses may reflect the activation of the JAK-STAT pathway in both bystander and infected cells.

The cellular RNA found to be the most significantly differentially methylated in response to SARS-CoV-2 infection was the precursor of miR-4486 (Table 2 and 3 and Fig. 2B). To our knowledge, an association between miR-4486 and viral infection has not previously been reported. Due to the size of the microarray probes (60 nucleotides [nt]), miR transcripts are represented by the unprocessed primary and precursor transcripts, which may serve as a proxy for the mature 22-nt functional miR. While the epitranscriptomic array showed an ~175-fold increase of m⁶A-modified pre-miR-4486 in infected cells, qPCR analysis only resulted in an ~22-fold enrichment of mature miR-4486 (Fig. 3D). This discrepancy could simply be due to the use of replication-competent virus for epitranscriptomic array versus the single-cycle replicon used for validation. However, m⁶A modification of miRNA precursors may promote their processing to mature miRNA (43). Therefore, the enrichment of m⁶A-modified miR-4486 in infected cells may simply reflect the more efficient processing of the methylated precursor.

One functionally validated target of miR-4486 is JAK3, which was also identified in our analysis as an upstream regulator significantly associated with differentially methylated transcripts in SARS-CoV-2-infected cells (Fig. 5 and 6C) (44). One possible hypothesis based on our network analysis is that degradation of JAK3 transcript by miR-4486 leads to lower STAT3 activation as a compensatory mechanism in the infected cell to counteract SARS-CoV-2-induced STAT3 hyperactivation (Fig. 6C) (45). The cellular lncRNA SNHG20 acts as a competing endogenous RNA to sponge miR-4486 and prevents the degradation of miR-4486 target transcripts (46, 47). Interestingly, a meta-analysis of transcriptomic data sets from COVID-19 patient samples found that SNHG20 was among the top 10 most significantly upregulated lncRNAs (48). It is possible that miR-4486 is upregulated early in infected cells, whereas SNHG20 is upregulated at later times to counteract the inhibition of cellular gene expression by miR-4486. Finally, experimentally validated binding targets of miR-4486 include TNF receptor-associated factor 7 (TRAF7) (49) and interleukin-1 receptor-associated kinase 3 (IRAK3) (49), both of which are involved in innate immune signaling and can lead to inhibition of NF-κB activation (50, 51). Further studies are needed to determine which of these miR-4486 targets are functionally relevant during SARS-CoV-2 infection of human lung epithelial cells.

EGFR was identified as the putative upstream regulator predicted to be associated with differentially methylated mRNA transcripts with the highest confidence (Fig. 5). A network map illustrates the potential functional interactions between EGFR and proteins downstream whose mRNA was found to be hypermethylated (RAS p21 protein activator 1 [RASA1] and STAT3) and hypomethylated (hypoxia-inducible factor 1 [HIF1A] and BCL2-like 11 [BCL2L11]). Several studies have demonstrated that EGFR is highly expressed and that EGFR signaling activity contributes to lung fibrosis in COVID-19 patients, leading to the identification of EGFR as a potential therapeutic target for treating severe COVID-19 (33, 35, 36).

The effect of reversible and dynamic m⁶A modification on a given transcript is context dependent and may be inhibitory (destabilization or sequestration) or activating (enhanced translation or splicing) (7–12). Networks such as those shown in Fig. 6 are useful for predicting functional associations between differentially methylated transcripts and identifying other members of a given signaling pathway that may also be affected by the change in m⁶A modification. However, due to the potential opposing functional effects of m⁶A modification on an mRNA, it may not be straightforward to predict the directionality of effect on a downstream pathway member caused by changes in methylation of an upstream mRNA target.

A limitation of our study is the use of a single human lung epithelial cell line. Our future studies are focused on the functional validation of selected differentially methylated transcripts. Specifically, it will be interesting to determine the biological significance of transcript methylation status during SARS-CoV-2 infection of differentiated primary human airway epithelial cells grown at air-liquid interface to more authentically mimic physiological conditions in the lung. Another limitation is that our analyses were carried out at a single time postinfection. It may also be informative to assess

changes in cellular transcript methylation using synchronized infection and analysis of different phases of the viral life cycle.

In summary, our analysis identifies many cellular RNAs that exhibit differential methylation in SARS-CoV-2-infected A549-hACE2 cells. These results can lay the foundation for the broader research community for the formation of novel hypotheses regarding the role of posttranscriptional regulation of host gene expression during SARS-CoV-2 infection.

## MATERIALS AND METHODS

**Cells and SARS-CoV-2 infection.** A549-hACE2 lung carcinoma cells expressing the human ACE2 protein (Invivogen) were maintained in DMEM with 4.5 g/L glucose and 2 mM L-glutamine (Gibco), 10% heat-inactivated fetal bovine serum (FBS; R&D Systems), 100 U/mL penicillin, 100 $\mu$g/mL streptomycin (Gibco), and 0.5 $\mu$g/mL puromycin (Sigma). Vero E6 TMPRSS2 cells were maintained in DMEM with 4.5 g/L glucose and 2 mM L-glutamine (Gibco), 10% heat-inactivated FBS (R&D Systems), 100 U/mL penicillin, and 100 $\mu$g/mL streptomycin (Gibco), and 5 $\mu$g/mL blasticidin (Invivogen). SARS-CoV-2 strain 2019n-CoV/USA-WA-1/2020 (BEI; cat. no. NR-52281) was propagated on Vero TMPRSS2 cells as described previously (52). Infected cell supernatant containing virus was collected, passed through a 0.45-$\mu$M filter, and concentrated by centrifugation at 10,000 $\times$ $g$ for 24 h at 10°C. Virus titer was determined by TCID$_{50}$ assay on Vero TMPRSS2 cells (52). ΔS-VRP(G) particles were generated as described previously (28). WT SARS-CoV-2 or ΔS-VRP(G) particles were used to infect cells at 37°C for 2 h. Fresh medium was then added to the infected cultures, followed by incubation at 37°C for an additional 24 h before RNA harvest.

**Immunofluorescence.** At 24 h postinfection, A549-hACE2 cells were fixed with 4% PFA (Electron Microscopy Sciences; cat. no. 15710) for 30 min at room temperature and permeabilized with 0.5% Triton X-100 in phosphate-buffered saline. Cells were incubated with rabbit monoclonal SARS-CoV/SARS-CoV-2 nucleocapsid antibody (SinoBiological 40143-R001; dilution 1:100), followed by incubation with Alexa Fluor 488 goat anti-rabbit IgG (Life Technologies; dilution 1:500). Nuclei were stained with DAPI (1 $\mu$g/mL). Images were acquired with a Nikon Eclipse Ts2 microscope.

**RT-qPCR.** Total cellular RNA was purified using TRIzol reagent according to the manufacturer's protocol (Invitrogen). RNA concentration was determined using a Nanodrop-OneC spectrophotometer (Thermo Fisher). RNA was DNase-treated using TURBO DNase according to the manufacturer's protocol (Thermo). DNase-treated RNA (100 ng) was used as the template for cDNA synthesis using the iScript cDNA Synthesis kit according to the manufacturer's protocol (Bio-Rad). The resulting cDNA was diluted 1:10 and 2 $\mu$L was used for qPCR amplification of SARS-CoV-2 spike using iTaq Universal SYBR Green Supermix according to the manufacturer's protocol (Bio-Rad). Copy number was calculated in reference to a standard curve of known copy number ($10^2$ to $10^7$ copies spike). Sequences of all primers used for qPCR amplification in this study are provided in Table S3. RT-qPCR detection of miR-4486 was performed using TaqMan miRNA assay ID 465336_mat according to the manufacturer's protocol (Thermo Fisher).

**m⁶A immunoprecipitation.** Total cellular RNA was purified using TRIzol reagent according to the manufacturer's protocol (Invitrogen). RNA concentration was determined using a Nanodrop-OneC spectrophotometer (Thermo Fisher). Five micrograms of total RNA and m⁶A spike-in control mixture was added to 300 $\mu$L 1$\times$ IP buffer (50 mM Tris-HCl, pH 7.4, 150 mM NaCl, 0.1% NP-40, and 40 U/$\mu$L RNase inhibitor) containing 2 $\mu$g anti-m⁶A rabbit polyclonal antibody (Synaptic Systems). The reaction mixture was incubated with head-over-tail rotation at 4°C for 2 h. Twenty microliters of Dynabeads M-280 sheep anti-rabbit IgG suspension per sample was blocked with freshly prepared 0.5% BSA at 4°C for 2 h, washed three times with 300 $\mu$L 1$\times$ IP buffer, and resuspended in the total RNA-antibody mixture prepared above. The RNA binding to the m⁶A-antibody beads was carried out with head-over-tail rotation at 4°C for 2 h. The beads were then washed three times with 500 $\mu$L 1$\times$ IP buffer and twice with 500 $\mu$L wash buffer (50 mM Tris-HCl, pH 7.4, 50 mM NaCl, 0.1% NP-40, and 40 U/$\mu$L RNase inhibitor). The enriched RNA was eluted with 200 $\mu$L elution buffer (10 mM Tris-HCl, pH 7.4, 1 mM EDTA, 0.05% SDS, 40 U proteinase K, and 1 $\mu$L RNase inhibitor) at 50°C for 1 h. The RNA was extracted by acid phenol-chloroform and ethanol precipitated. For RT-qPCR analysis, all RNA purified from a single m⁶A-RIP was used for cDNA synthesis. Data were analyzed by the $\Delta\Delta C_T$ method using a spike-in luciferase RNA as an internal control (New England Biolabs).

**RNA labeling and hybridization.** The IP RNAs and unbound RNAs were mixed with an equal amount of calibration spike-in control RNA, separately amplified, and labeled with Cy3 (unbound) and Cy5 (IP) using Arraystar Super RNA Labeling kit. The synthesized cRNAs were purified by RNeasy minikit (Qiagen). The concentration and specific activity (pmol dye/$\mu$g cRNA) were measured with NanoDrop ND-1000. Cy3- and Cy5-labeled cRNAs were mixed, and 2.5 $\mu$g of the cRNA mixture in a 19-$\mu$L volume was fragmented by adding 5 $\mu$L 10$\times$ blocking agent and 1 $\mu$L of 25$\times$ fragmentation buffer, followed by heating at 60°C for 30 min. The fragmented RNA was combined with 25 $\mu$L 2$\times$ hybridization buffer. Then, 50 $\mu$L hybridization solution was dispensed into the gasket slide and assembled to the m⁶A-mRNA&lncRNA Epitranscriptomic Microarray slide. The slides were incubated at 65°C for 17 h in an Agilent hybridization oven. The hybridized arrays were washed, fixed, and scanned using an Agilent scanner (G2505C).

**Data and statistical analysis.** Agilent Feature Extraction software (version 11.0.1.1) was used to analyze acquired array images. Raw intensities of IP (Cy5 labeled) and unbound (Cy3 labeled) were normalized using the average of log$_2$-scaled spike-in RNA intensities. After spike-in normalization, the probe signals having present or marginal quality control flags in at least three out of six samples were retained for further

"m⁶A quantity" analyses. "m⁶A quantity" was calculated for the m⁶A methylation amount based on the IP (Cy5-labeled) normalized intensities. Differentially m⁶A-methylated RNAs between two comparison groups were identified by filtering with the fold change and statistical significance ($P$ value) thresholds. Protein-coding transcripts that were found to be differentially m⁶A methylated between mock- and SARS-CoV-2-infected samples with a fold change ≥1.5 and $P \leq 0.05$ were uploaded to iPathwayGuide (ipathwayguide .advaitabio.com) for gene ontology, pathway, upstream regulator, and network analyses (29–32).

## SUPPLEMENTAL MATERIAL

Supplemental material is available online only.

**SUPPLEMENTAL FILE 1**, XLSX file, 0.1 MB.
**SUPPLEMENTAL FILE 2**, XLSX file, 0.02 MB.
**SUPPLEMENTAL FILE 3**, XLSX file, 0.01 MB.
**SUPPLEMENTAL FILE 4**, PDF file, 0.02 MB.

## ACKNOWLEDGMENTS

We thank Michael Chimenti for helpful discussion and assistance with iPathway Guide analysis. We thank Patrick Sinn for A549-hACE2 cells. We thank Balaji Manicassamy for providing us with the SARS-CoV-2 replicon plasmid (ΔS Luc-GFP). We declare that they have no known competing financial interests or personal relationships that could have appeared to influence the work reported in this paper.

This work was supported by the National Institutes of Health (NIH) grant R21AI159546 to L.W. and grant R01AI134733 to W.M. The funder had no role in study design, data collection, and analysis, decision to publish, or preparation of the manuscript. The following reagent was deposited by the Centers for Disease Control and Prevention and obtained through BEI Resources, NIAID, NIH: SARS-Related Coronavirus 2, Isolate USA-WA1/2020, NR-52281.

Conceptualization, Methodology, Formal analysis, Investigation, Writing – Original Draft, Writing – Review & Editing, Visualization, Supervision, Project Administration, Stacia Phillips. Validation, Formal analysis, Investigation, Tarun Mishra. Investigation, Shaubhagya Khadka. Investigation, Writing – Review & Editing, Constanza E. Espada. Investigation, Writing – Review & Editing, Dana Bohan. Methodology, Resources, Writing – Review & Editing, Supervision, Wendy Maury. Conceptualization, Methodology, Writing – Review & Editing, Visualization, Supervision, Project Administration, Funding Acquisition, Li Wu.

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
