## [Reviewer comments · Microbiology Spectrum]

Microbiology Spectrum

Epitranscriptomic N⁶-methyladenosine profile of SARS-CoV-2-infected human lung epithelial cells

Stacia Phillips, Tarun Mishra, Shaubhagya Khadka, Dana Bohan, Constanza Espada, Wendy Maury, and Li Wu

Corresponding Author(s): Li Wu, The University of Iowa

Review Timeline:

Submission Date:	September 27, 2022
Editorial Decision:	December 5, 2022
Revision Received:	December 9, 2022
Accepted:	December 18, 2022

Editor: Yongjun Sui

Reviewer(s): The reviewers have opted to remain anonymous.

Transaction Report:

DOI: <https://doi.org/10.1128/spectrum.03943-22>

December 5, 2022

Dr. Li Wu
The University of Iowa
Microbiology and Immunology
51 Newton Road
Iowa City 52242

Re: Spectrum03943-22 (Epitranscriptomic N⁶-methyladenosine profile of SARS-CoV-2-infected human lung epithelial cells)

Dear Dr. Li Wu:

Link Not Available

Sincerely,

Yongjun Sui

Journals Department
Reviewer comments:

Reviewer #1 (Comments for the Author):

Phillips et al present data to define the epitranscriptomic m6A profile of SARS-CoV-2-infected human lung epithelial cells and draw comparison to uninfected control cells. The study identified mRNA, LncRNA and sncRNA species that are differentially m6A-modified in response to SARS-CoV-2 infection. Interestingly, expression of m6A-modified miRNA-4486 was significantly upregulated in SARS-CoV-2-infected cells. This is a very intriguing finding to report even though the study does not provide functional consequence of miRNA-4486 upregulation. However, a reasonable discussion is provided.

Some important suggestions for authors to consider:

- 1) Could the study provide a comparison of differentially expressed coding and non-coding RNAs between their study and the published SARS-CoV-2 studies with respect to m6A-modified RNA species (such as reference 14).
- 2) Can authors report whether miRNA-4486 upregulation has been reported for other CoVs (or other viruses in general)?
- 3) Does SARS-CoV-2 infection upregulate miRNA-4486 expression in primary airway epithelial cells (HBEs/HAEs) of human, mice, or NHP origin?
- 4) Is non-m6A-modified miRNA-4486 also upregulated during SARS-CoV-2 infection? Or is there selective upregulation of m6A-modified miRNA-4486?

Reviewer #2 (Comments for the Author):

Here, Phillips et al. describe changes to post-transcriptional modification of host RNA in cultured lung cells triggered by infection with the virus sudden acute respiratory syndrome coronavirus type 2 (SARS-CoV-2) that causes COVID-19. The focus is N6-methyladenosine (m6A) modifications that are commonly triggered by viral infection and may affect RNA stability or utilization relevant to mRNA translation or other regulatory processes. The study is straightforward- the authors isolate RNA from uninfected and infected A549 cells at 24 hours post-infection prior to m6A-immunoprecipitation for enrichment followed by commercial microarray to measure changes to relative m6A-modified transcript abundance. Broad changes to the host m6A "epitranscriptome" are identified. Most are moderate but with one particular hit standing out- m6A-modified miRNA-4486, enriched >100-fold relative to uninfected cells. Pathway analyses indicate several genes, e.g., MUSK, STAT3, and EGFR targets as having potential for future investigation. The paper is very cleanly written and the techniques and analytical tools are appropriate. The dataset seems robust and will be of use to the field, fitting the scope of the journal. The major limitation, however, is that only a single cell line (A549) is studied. Validation of these changes, in particular the compelling miRNA results, would seem warranted in at least one additional cell system.

Major comments:

1. As noted above, the major limitation to the study is that it only features a single infected cell type at a single time point post-infection. Specifically, it seems important to confirm or deny that miRNA-4486 and other hits are common targets of SARS-CoV-2 infection.

Minor comments:

1. The authors' abstract claim that their data "...suggest that m6A modification of cellular RNA is an important mechanism regulating host gene expression during SARS-CoV-2 infection..." is not justified herein. Changes are observed and reported consistent with the scope of this journal. Mechanism and/or relevance should be either addressed experimentally or this type of language removed.
2. Hits (and misses) could also be better discussed (i.e., metanalysis) in the context of numerous competing studies (note refs. 14-18).
3. Figure 2A. Something about this scheme confused me- I think it is that the shifted positions of transcript pools A and B implied some sort of interaction between the two pools.
4. Figure 2B. The 4486 dot was not distinguishable in my version (all appear red to me, but could be that I am partially color-blind.). I would also recommend highlighting the other notable hits and what they are so that is less necessary to access the supplementary data.
5. Table 2. Please better explain the criteria for selection of genes being shown here and the ordering- seems more useful for readers to go straight to the spreadsheets where they can sort the data, table feels arbitrary (i.e., what is its utility?).

Staff Comments:

Preparing Revision Guidelines

To submit your modified manuscript, log onto the eJP submission site at <https://spectrum.msubmit.net/cgi-bin/main.plex>. Go to Author Tasks and click the appropriate manuscript title to begin the revision process. The information that you entered when you first submitted the paper will be displayed. Please update the information as necessary. Here are a few examples of required

updates that authors must address:

Please return the manuscript within 60 days; if you cannot complete the modification within this time period, please contact me. If you do not wish to modify the manuscript and prefer to submit it to another journal, please notify me of your decision immediately so that the manuscript may be formally withdrawn from consideration by Microbiology Spectrum.

Point-by-point responses to reviewer comments:

Spectrum03943-22

Phillips *et al.* Epitranscriptomic N^6 -methyladenosine profile of SARS-CoV-2-infected human lung epithelial cells.

Reviewer #1:

Phillips et al present data to define the epitranscriptomic m6A profile of SARS-CoV-2-infected human lung epithelial cells and draw comparison to uninfected control cells. The study identified mRNA, LncRNA and sncRNA species that are differentially m6A-modified in response to SARS-CoV-2 infection. Interestingly, expression of m6A-modified miRNA-4486 was significantly upregulated in SARS-CoV-2-infected cells. This is a very intriguing finding to report even though the study does not provide functional consequence of miRNA-4486 upregulation. However, a reasonable discussion is provided.

Some important suggestions for authors to consider:

1) Could the study provide a comparison of differentially expressed coding and non-coding RNAs between their study and the published SARS-CoV-2 studies with respect to m6A-modified RNA species (such as reference 14).

Author response: We agree that a comparison of differential m⁶A-modification across studies would be beneficial for our broader understanding of how m⁶A modification of host cell transcripts change in response to infection. Liu *et al.* (cited reference 14) reported “gain and loss” of m⁶A peaks from sequencing results derived from m⁶A-modified mRNA in SARS-CoV-2-infected cells relative to mock controls. However, there are important differences in experimental design and methodology that make meaningful comparison of our results difficult:

A) Liu *et al.* used Huh7 (hepatocyte-derived carcinoma cells) infected at an MOI of 0.05 for 120 hours. We used A549-hACE2 (lung-derived carcinoma cells) infected at an MOI of 1.0 for 24 hours.

B) Liu *et al.* performed meRIP-seq to identify m⁶A-modified sites in cellular RNA. This method requires the fragmentation of RNA prior to m⁶A-IP. As a result, transcripts may have both gain and loss of peaks on the same RNA, depending on the exact site of modification. In contrast, the epitranscriptomic array utilized in our study does not involve RNA fragmentation. Therefore, our analyses will reveal a net overall increase or decrease in m⁶A modification, without information regarding site-specificity. While there is some overlap in the two data sets, we chose to focus our discussion on our results.

2) Can authors report whether miRNA-4486 upregulation has been reported for other CoVs (or other viruses in general)?

Author response: To our knowledge, miR-4486 upregulation has not previously been implicated in the context of viral infection. We have added a comment reflecting such to the discussion (lines 261-263).

3) Does SARS-CoV-2 infection upregulate miRNA-4486 expression in primary airway epithelial cells (HBEs/HAEs) of human, mice, or NHP origin?

Author response: Like the reviewer, we are interested to know whether m⁶A modification of miRNA-4486 is upregulated in SARS-CoV-2-infected primary HBE or HAE culture. While these experiments are outside the scope of our current study, we have added language to the text (lines 304-308) to indicate that this will be an important area of future investigation.

4) Is non-m⁶A-modified miRNA-4486 also upregulated during SARS-CoV-2 infection? Or is there selective upregulation of m⁶A-modified miRNA-4486?

Author response: We appreciate the reviewer raising this important point. We have clarified the results in the text (lines 149-152) and with additional data (Fig. 3C-D).

Reviewer #2:

Here, Phillips et al. describe changes to post-transcriptional modification of host RNA in cultured lung cells triggered by infection with the virus sudden acute respiratory syndrome coronavirus type 2 (SARS-CoV-2) that causes COVID-19. The focus is N⁶-methyladenosine (m⁶A) modifications that are commonly triggered by viral infection and may affect RNA stability or utilization relevant to mRNA translation or other regulatory processes. The study is straightforward- the authors isolate RNA from uninfected and infected A549 cells at 24 hours post-infection prior to m⁶A-immunoprecipitation for enrichment followed by commercial microarray to measure changes to relative m⁶A-modified transcript abundance. Broad changes to the host m⁶A "epitranscriptome" are identified. Most are moderate but with one particular hit standing out- m⁶A-modified miRNA-4486, enriched >100-fold relative to uninfected cells. Pathway analyses indicate several genes, e.g., MUSK, STAT3, and EGFR targets as having potential for future investigation. The paper is very cleanly written and the techniques and analytical tools are appropriate. The dataset seems robust and will be of use to the field, fitting the scope of the journal. The major limitation, however, is that only a single cell line (A549) is studied. Validation of these changes, in particular the compelling miRNA results, would seem warranted in at least one additional cell system.

Major comments:

1. As noted above, the major limitation to the study is that it only features a single infected cell type at a single time point post-infection. Specifically, it seems important to confirm or deny that miRNA-4486 and other hits are common targets of SARS-CoV-2 infection.

Author response: We agree that our study would be strengthened by comparing results from other cell types or times post-infection. Due to the cost associated with the epitranscriptomic array, we chose to specifically focus on a lung cell line, which we feel is more biologically relevant to SARS-CoV-2 infection than other commonly used cell types in the field, such as Vero or Huh7. We strategically chose 24 hours post-infection because our data (not shown) shows that this represents a spreading infection, and therefore should capture modified RNA representing each temporal phase of the viral life cycle. We have added text (lines 304-311) to make clear that this is a limitation of our study.

Minor comments:

1. The authors' abstract claim that their data "...suggest that m⁶A modification of cellular RNA is an important mechanism regulating host gene expression during SARS-CoV-2 infection..." is not justified herein. Changes are observed and reported consistent with the scope of this journal. Mechanism and/or relevance should be either addressed experimentally or this type of language removed.

Author response: We agree with the reviewer that this language was an overreach and not supported by our data. It has been removed.

2. Hits (and misses) could also be better discussed (i.e., metanalysis) in the context of numerous competing studies (note refs. 14-18).

Author response: We thank the reviewer for this suggestion. However, the citations referred to above as competing studies focus on m⁶A modification of viral RNA, with little or no reference to changes in the modification state of cellular RNA. The only study that reported m⁶A changes in cellular RNA at the transcript level is Liu *et al.* For reasons discussed above in response to reviewer 1 (comment 1), we chose to provide a more focused discussion without making comparisons that the preceding study.

3. Figure 2A. Something about this scheme confused me- I think it is that the shifted positions of transcript pools A and B implied some sort of interaction between the two pools.

Author response: We appreciate the reviewer's comment. We have edited Figure 2A to remove this potentially confusing layout.

4. Figure 2B. The 4486 dot was not distinguishable in my version (all appear red to me, but could be that I am partially color-blind.). I would also recommend highlighting the other notable hits and what they are so that is less necessary to access the supplementary data.

Author response: We appreciate the reviewer's comment. We have edited Figure 2B to label specific transcripts.

5. Table 2. Please better explain the criteria for selection of genes being shown here and the ordering- seems more useful for readers to go straight to the spreadsheets where they can sort the data, table feels arbitrary (i.e., what is its utility?).

Author response: The selection criteria for Table 2 is explained in the text (lines 122-125). We think that the inclusion of tables in the main body of the manuscript to summarize the most significantly changed transcripts is useful for readers who may not wish to access the supplemental Excel files.

December 18, 2022

Dr. Li Wu
The University of Iowa
Microbiology and Immunology
51 Newton Road
Iowa City 52242

Re: Spectrum03943-22R1 (Epitranscriptomic N⁶-methyladenosine profile of SARS-CoV-2-infected human lung epithelial cells)

Dear Dr. Li Wu:

Your manuscript has been accepted, and I am forwarding it to the ASM Journals Department for publication. You will be notified when your proofs are ready to be viewed.

Sincerely,

Yongjun Sui
Editor, Microbiology Spectrum

Journals Department
Supplemental Table S2: Accept
Supplemental Table S1: Accept
Supplemental Table S3: Accept
Supplemental table legends: Accept